# Loganin Attenuates the Severity of Acute Kidney Injury Induced by Cisplatin through the Inhibition of ERK Activation in Mice

**DOI:** 10.3390/ijms22031421

**Published:** 2021-01-31

**Authors:** Dong-Uk Kim, Dong-Gu Kim, Ji-Won Choi, Joon Yeon Shin, Bitna Kweon, Ziqi Zhou, Ho-Sub Lee, Ho-Joon Song, Gi-Sang Bae, Sung-Joo Park

**Affiliations:** 1Hanbang Cardio-Renal Syndrome Research Center, Wonkwang University, Iksan, Jeonbuk 54538, Korea; ckck202@hanmail.net (D.-U.K.); kdg2409@naver.com (D.-G.K.); agraesb1@naver.com (J.-W.C.); host@wku.ac.kr (H.-S.L.); 2Department of Herbology, School of Korean Medicine, Wonkwang University, Iksan, Jeonbuk 54538, Korea; joonyeon3@naver.com (J.Y.S.); kbn306@naver.com (B.K.); zzq0325@naver.com (Z.Z.); songhj@wku.ac.kr (H.-J.S.); 3Department of Herbal Resources, Professional Graduate School of Korean Medicine, Wonkwang University, Iksan, Jeonbuk 54538, Korea; 4Department of Pharmacology, School of Korean Medicine, Wonkwang University, Iksan, Jeonbuk 54538, Korea; 5Research Center of Traditional Korean Medicine, Wonkwang University, Iksan, Jeonbuk 54538, Korea

**Keywords:** acute kidney injury, apoptosis, cisplatin, erk, ferroptosis, loganin

## Abstract

Cisplatin is the most widely used chemotherapeutic agent. However, it often causes nephrotoxicity, which results in acute kidney injury (AKI). Therefore, we urgently need a drug that can reduce the nephrotoxicity induced by cisplatin. Loganin is a major iridoid glycoside isolated from Corni fructus that has been used as an anti-inflammatory agent in various pathological models. However, the renal protective activity of loganin remains unclear. In this study, to examine the protective effect of loganin on cisplatin-induced AKI, male C57BL/6 mice were orally administered with loganin (1, 10, and 20 mg/kg) 1 h before intraperitoneal injection of cisplatin (10 mg/kg) and sacrificed at three days after the injection. The administration of loganin inhibited the elevation of blood urea nitrogen (BUN) and creatinine (CREA) in serum, which are used as biomarkers of AKI. Moreover, histological kidney injury, proximal tubule damages, and renal cell death, such as apoptosis and ferroptosis, were reduced by loganin treatment. Also, pro-inflammatory cytokines, such as interleukin (IL)-1β, IL-6, and tumor necrosis factor (TNF)-α, reduced by loganin treatment. Furthermore, loganin deactivated the extracellular signal-regulated kinases (ERK) 1 and 2 during AKI. Taken together, our results suggest that loganin may attenuate cisplatin-induced AKI through the inhibition of ERK1/2.

## 1. Introduction

Cisplatin is widely used for the treatment of malignant tumors in the head and neck, lungs, ovaries, and the bladder [1]. The cytotoxic activity of cisplatin against tumor cells mainly involves cross-linked purine bases within the DNA, and it interferes with DNA synthesis and induces apoptosis [2]. However, cisplatin induces several side effects, such as hepatotoxicity [3,4], cardiotoxicity [5,6], gastrotoxicity [7], ototoxicity [8,9], and nephrotoxicity [10,11,12]. Nephrotoxicity is a frequent and major complication of cisplatin-based chemotherapy; it can cause acute kidney injury (AKI) mediated by apoptosis of renal proximal tubule cells [12]. When AKI occurs, blood urea nitrogen (BUN), serum creatinine (CREA), and the glomerular filtration rate are elevated, which ultimately leads to various kidney diseases, such as hypomagnesemia, hypocalcemia, renal salt wasting, renal concentrating defect, and hyperuricemia [13,14].

Loganin is an iridoid glycoside found in Corni fructus, which has been used to replenish the liver and kidney and suppress sweat and seminal emission [15]. Previously, we reported the beneficial role of loganin in acute pancreatitis [16], and several studies have reported that loganin has anti-inflammatory [17], anti-diabetic [18], anti-apoptotic [19], and anti-shock properties [20]. In addition, the metabolite compound of loganin is reported to have anti-tumor effects on gastric cancer cells and synergistic effects on 5-fluorouracil-induced cell death [21].

However, the effect of loganin on cisplatin-induced AKI has not been reported. In this study, we investigated the effect of loganin on cisplatin-induced AKI in mice. Bodyweight, serum BUN and CREA levels, histological appearance of the kidney, renal cell death, and pro-inflammatory cytokines were evaluated to determine the protective effect of loganin against cisplatin-induced AKI. Furthermore, we investigated the inhibitory mechanisms of loganin by evaluating the activation of mitogen-activated protein kinases (MAPKs) and nuclear factor kappa B (NF-κB).

## 2. Results

### 2.1. Effects of Loganin on Body Weight and Serum Biochemical Markers in Cisplatin-Induced AKI

To examine the effects of loganin on AKI, the mice were administered with loganin 1 h before cisplatin injection and sacrificed at 72 h after cisplatin injection (Figure 1B). We found that cisplatin treatment caused body weight loss and elevated levels of serum biochemical parameters such as BUN and CREA. However, oral administration of loganin demonstrated dose-dependent inhibition of these effects (Figure 1C,D, Table 1).

### 2.2. Effects of Loganin on Renal Injury in Cisplatin-Induced AKI

Cisplatin caused histological damage, such as necrosis of tubular cells. However, loganin treatment significantly alleviated the renal histological damage in a dose-dependent manner (Figure 2A,C). To assess the injury of proximal renal tubule cells, we detected the brush border with immunofluorescence staining of lotus tetragonolobus lectin (LTL). Treatment with cisplatin only caused a loss of LTL-positive cells. On the other hand, loganin treatment restored the loss of LTL-positive proximal renal tubule cells in cisplatin-induced AKI (Figure 2B,D).

### 2.3. Effects of Loganin on Renal Cell Death, such as Apoptosis, Ferroptosis, and Pyroptosis in Cisplatin-Induced AKI

Apoptosis in renal tubules was assessed using the terminal deoxynucleotidyl transferase dUTP nick-end labeling (TUNEL) assay. TUNEL-positive cells were almost undetectable in the control group, while the number of the TUNEL-positive cells was increased after cisplatin administration. However, loganin treatment significantly reduced the TUNEL-positive cells in a dose-dependent manner (Figure 3A). Next, to detect renal cell ferroptosis in AKI, we examined anti-4-hydroxynonenal (4-HNE) as the marker of lipid peroxidation and glutathione peroxidase 4 (GPX4) as the central modulator of ferroptosis. We found that 4-HNE was significantly increased, while GPX4 was decreased by cisplatin, which suggests that ferroptosis occurred during AKI. However, the increase of 4-HNE and the decrease of GPX4 were reversed by loganin treatment (Figure 3C,D). Furthermore, we also examined whether the pyroptosis in renal tubules was inhibited by loganin during AKI. Pyroptosis was detected by formation of NACHT, LRR, and PYD domains-containing protein 3 (NLRP3) inflammasome complex and activation of caspase-1 and gasdermin D (GSDMD), and these typical markers for pyroptosis were increased in cisplatin-induced AKI mice. However, loganin treatment inhibited the inflammasome complex (NLRP3 and PYCARD) and cleavage of caspase-1 but not GSDMD (Appendix A).

### 2.4. Effects of Loganin on Renal Cytokines Production in Cisplatin-Induced AKI

Many studies have reported that pro-inflammatory cytokines, including interleukin (IL)-1β, IL-6, and tumor necrosis factor (TNF)-α, were increased during cisplatin-induced AKI. Hence, we examined the renal mRNA levels of IL-1β, IL-6, and TNF-α. As shown in Figure 4, these pro-inflammatory cytokines were increased in cisplatin-induced AKI mice. However, loganin treatment inhibited the mRNA levels of IL-1β, IL-6, and TNF-α in a dose-dependent manner.

### 2.5. Effects of Loganin on the Activation of Extracellular Signal-Regulated Kinases (ERK) 1/2 in Cisplatin-Induced AKI

To investigate the inhibitory mechanisms of loganin against cisplatin-induced AKI, the activation of mitogen activated protein kinases (MAPKs) and nuclear factor (NF)-κB was examined in the kidney. Cisplatin treatment triggered the phosphorylation of MAPKs and the degradation of inhibitory κ-Bα (Iκ-Bα) in the kidney. However, the administration of loganin inhibited the phosphorylation of ERK 1/2 only, not the phosphorylation p38 and c-Jun *N*-terminal kinase (JNK) and the degradation of Iκ-Bα (Figure 5A,B and Appendix A). Consistent with Figure 5A, ERK 1/2 phosphorylation was also inhibited by loganin on immunofluorescence staining (Figure 5C).

### 2.6. Effects of ERK 1/2 on the Severity in Cisplatin-Induced AKI

To examine whether the inhibition of ERK activation could improve cisplatin-induced AKI, the mice were administered with U0126 (the well-known ERK 1/2 inhibitor) 1 h before cisplatin injection and sacrificed at 72 h after cisplatin injection. We found that the inhibition of ERK 1/2 by U0126 suppressed not only histological damage by cisplatin but also the increase of serum BUN and CREA levels and inflammatory cytokines (Figure 6).

## 3. Discussion

Cisplatin is a clinically effective anti-cancer agent, but it has serious complications, especially AKI [10,11,12]. To counteract the side effects of cisplatin, the development of new products or agents is urgently needed. In this study, we suggested loganin as a regulator of side effects induced by cisplatin and investigated the effect of loganin in a clinically relevant model of cisplatin-induced AKI. We showed that loganin attenuated the cisplatin-induced changes in BUN and CREA, histological injury, proximal tubule cell death, and pro-inflammatory cytokine production. Our results suggest that loganin can attenuate cisplatin-induced AKI. In addition, we explored the underlying molecular mechanisms and found that loganin inhibited the cisplatin-induced activation of ERK 1/2 in the kidney, and the inhibition of ERK 1/2 inhibited the severity of cisplatin-induced AKI. These findings support the potential attenuation of cisplatin-induced AKI by loganin via the deactivation of ERK 1/2.

In this study, we tried to examine the role of loganin against AKI using a concentration of less than 20 mg/kg. Based on our previous reports [16], we had information that loganin was not toxic up to 100 mg/kg by oral administration in mice. Thus, we firstly conducted the experiments with loganin concentrations of 20, 50, and 100 mg/kg. In these preliminary experiments (20, 50, and 100 m/kg of loganin), 20 mg/kg of loganin improved the severity of cisplatin-induced AKI; however, 50 and 100 mg/kg of loganin did not show any significant improvement (Appendix A). Therefore, we decided to lower the concentration range of loganin to less than 20 mg/kg and then performed the experiment using the 1, 10, and 20 mg/kg of loganin reported in this study.

Cisplatin-induced AKI involves a complex multi-process [22]. Firstly, cisplatin directly causes the toxicity of the renal proximal tubular cells and renal blood vessels, resulting in a decline in renal blood flow and glomerular filtration rate. Consequently, renal function declines rapidly, resulting in the accumulation of waste products such as creatinine and urea [23]. Secondly, cisplatin activates various inflammatory mediators such as cytokines and signaling molecules, which causes irreversible renal damage [11]. Therefore, we evaluated the two processes to determine the effects of loganin during cisplatin-induced AKI in this study.

Firstly, to examine whether loganin protects against the first cascade of cisplatin-induced AKI, we investigated renal function and renal tubule cell damage. To investigate renal function, we examined the serum levels of BUN and CREA to demonstrate renal physiological function during AKI [24]. Consistent with previous reports [25,26], our results show that serum BUN and CREA were significantly elevated in the cisplatin-induced AKI model. However, the elevation of BUN and CREA was inhibited by loganin administration, which means that loganin may prevent the decline of renal function in cisplatin-induced AKI. To examine whether loganin attenuates the damage of renal proximal tubular cells, we investigated the necrosis of tubular cells, the loss of brush border, and renal cell apoptosis. We found necrosis of tubular cells (indicated by the histological picture), the loss of brush border (indicated by LTL staining), and renal cell apoptosis (indicated by TUNEL) in the cisplatin-induced AKI model (Figure 2 and Figure 3), which means cisplatin caused toxicity of the renal proximal tubular cells. However, necrosis and loss of brush border were inhibited by loganin treatment, which suggests that loganin protected against renal damage in the cisplatin-induced AKI model.

Although renal cell apoptosis is a well-known cell death type during AKI [27,28], various types of cell death such as ferroptosis and pyroptosis could have occurred [29,30]. Ferroptosis is a recently discovered cell death pathway that is characterized by an accompanying accumulation of iron ions and lipid peroxide [31,32]. The accumulated iron ions produce a large amount of lipid reactive oxygen species (ROS) that lead to lipid peroxidation and ferropotosis. Thus, upon ferroptosis-induced AKI, GPX4, an enzyme that negatively regulates lipid ROS, is decreased, and 4-HNE, a well-known byproduct of lipid peroxidation, is increased. In this study, we found that GPX4 was significantly downregulated, and 4-HNE was upregulated during AKI, in accordance with previous reports [33,34], which suggests ferroptosis is well-established in our AKI model. However, loganin treatment reversed the downregulation of GPX4 and upregulation of 4-HNE, which means loganin could protect against ferroptosis in cisplatin-induced AKI (Figure 4). Another type of cell death, pyroptosis is an inflammatory form of programmed cell death [35,36]. The process is initiated by the formation of inflammasome (also known as a pyroptosome), and then activates caspases that contribute to the release of several pro-inflammatory cytokines and pore-forming protein gasdermin D (GSDMD) [36]. Formation of GSDMD-cleaved N-terminal (GSDMD-N) causes cell membrane rupture and release of cytokines such as IL-1β and IL-18 [37]. Thus, we examined the expression of inflammasome complex (NLRP3, PYCARD), caspase-1, and GSDMD to detect pyroptosis. In accordance with previous reports [37], we found an increase of inflammasome complex, caspase-1 cleavage, and GSDMD in cisplatin-induced AKI (Appendix A). However, treatment of loganin inhibited the elevation of NLRP3, PYCARD, and cleavage of caspase-1 but not GSDMD (Appendix A). Therefore, these results might suggest that treatment with loganin inhibits only inflammasome-induced inflammatory condition but not pyroptosis in cisplatin-induced AKI.

Secondly, to examine the cisplatin-induced inflammatory pathways (the second cascade of cisplatin-induced AKI), we investigated the expression of pro-inflammatory cytokines in the kidney. Several studies have focused on the importance of pro-inflammatory cytokines in the treatment of AKI [38,39]. The representative pro-inflammatory cytokines, such as IL-1β, IL-6, and TNF-α, are reported to increase in cisplatin-induced AKI [40,41,42]. Genetic or pharmacological inhibition of TNF-α reduced the severity of AKI and the expression of other pro-inflammatory cytokines [42]. In this experiment, mRNA levels of IL-1β, IL-6, and TNF-α increased in cisplatin-induced AKI, which is consistent with previous reports [43,44]. However, loganin treatment inhibited the increase in pro-inflammatory cytokines, which suggests that loganin protects against renal damage via the inhibition of cisplatin-induced inflammatory responses (Figure 4).

After the release of pro-inflammatory cytokines, signaling molecules such as MAPKs and NF-κB are activated, which causes nephrotoxicity [45,46]. MAPKs and NF-κB pathways are well-known inflammatory cascades that regulate cell proliferation and death in several diseases [47]. Thus, we examined the activation of MAPKs and NF-κB in a cisplatin-induced AKI model. The administration of loganin inhibited the activation of ERK 1/2, but not JNK, p38, and NF-κB, and we assumed that the inhibition of ERK 1/2 signaling would be beneficial to reno-protection against cisplatin (Figure 5). ERK 1/2 signaling is reported to be associated with inflammation and cell death in AKI [25,48,49,50]. The inactivation of ERK 1/2 attenuated the severity of cisplatin-induced AKI by reducing the expression of pro-inflammatory cytokines, such as IL-1β, IL-6, and TNF-α [50]. In addition, we also showed that the administration of U0126, an ERK 1/2 inhibitor, improved the severity of cisplatin-induced AKI (Figure 6). This indicates that phosphorylation of ERK 1/2 is an upstream signal for inflammation and renal cell damage in cisplatin-induced AKI. Thus, we suggest that loganin improved the severity of cisplatin-induced AKI through the inhibition of ERK 1/2. In addition, the specific inhibitory effects of loganin on ERK 1/2 may facilitate a better understanding of effective pharmacological activities and help in the development of new drugs to treat AKI.

## 4. Materials and Methods

### 4.1. Chemicals and Reagents

Tris-HCl was purchased from Sigma-Aldrich (St. Louis, MO, USA). Easy-Blue total RNA extraction kit was purchased from iNtRON biotechnology (Sungnam, South Korea). Anti-4-hydroxynonenal (4-HNE) (1:500; ab48506), glutathione peroxidase 4 (GPX4) (1:500; ab125066), caspase-1 (1:500; ab1872) and gasdermin D (GSDMD) (1:500; ab219800) were purchased from Abcam (Cambridge, UK). Anti-caspase-1 (p20) (1:500; AG-20B-0042-C100) was purchased Adipogen (San Diego, CA, USA). Anti-phosphorylated extracellular signal-regulated kinase (ERK) 1/2 (1:1000; cat. no. 9101L), c-Jun *N*-terminal kinase (JNK) (1:1000; cat. no. 9251L), and p38 (1:1000; cat. no. 9211L) antibodies were purchased from Cell Signaling Technology (Danvers, MA, USA). Total MAPK antibodies against ERK 1/2 (1:1000; sc-93), JNK (1:1000; sc-474) and p38 (1:1000; sc-535), inhibitory κ-Bα (Iκ-Bα; 1:1000; sc-371), β-actin antibody (1:1000; sc-1615), horseradish peroxidase (HRP)-conjugated secondary antibody, including chicken anti-rabbit IgG-HRP (1:5000; sc-516087), and loganin were purchased from Santa Cruz Biotechnology (Dallas, TX, USA). Loganin stock solutions were prepared with distilled water.

### 4.2. Animal Models

All experiments were performed according to the protocols approved by the Animal Care Committee of Wonkwang University (WKU19 approved on October 2019). C57BL/6 mice (8–10 weeks old, male, weighing 20–25 g) were purchased from Orient Bio (Sungnam, South Korea). They were bred and housed in standard shoebox cages in a climate-controlled room with an ambient temperature of 23 ± 2 °C and a 12 h light–dark cycle for 7 days. They were also fed standard laboratory chow, provided with water ad libitum, and randomly assigned to control and experimental groups.

### 4.3. Experimental Design

AKI was induced by a single intraperitoneal injection of cisplatin (10 mg/kg). One h before cisplatin injection, mice in the loganin group received 1, 10, or 20 mg/kg of loganin orally, and mice in the U0126 group received 10 mg/kg of U0126 intraperitoneally. They were sacrificed at 72 h after cisplatin injection, and their blood and kidneys were collected. After separating the serum from the blood, it was immediately stored with the kidneys at −80 °C for further studies. To retain the refinement of mice, we measured the body weight and behavioral changes daily and excluded the mice with more than 25% of reduced behavioral activity or weight to follow the humane endpoints regarding experimental animals of AKI.

### 4.4. Measurement of Serum Biochemical Markers of AKI

Blood samples for the determination of serum BUN and CREA levels were obtained at 72 h after the injection of cisplatin. Mice were sacrificed via CO_2_ asphyxiation. The CO_2_ flow rate displaced 50% of the cage volume per minute. To ensure death following CO_2_ asphyxiation, cervical dislocation was also performed. Blood samples (about 0.5 mL) were then withdrawn from the heart. BUN and CREA were determined by an assay kit of Sekisui Medical (Tokyo, Japan).

### 4.5. Histological Analysis

After the mice were sacrificed, their kidneys were washed with ice-cold stroke-physiological saline solution and fixed in 10% neutral buffered formalin solution for 24 h. After routine processing, such as gradient alcohol dehydration and xylene permeabilization, the renal tissues were embedded in paraffin, sectioned at 4 μm thickness, and stained with periodic acid–Schiff (PAS; 0.5% periodic acid for 5 min and Schiff’s reagent for 15 min). The kidney sections representing a minimum of 100 fields of at least 3 mice from each group were semi-quantitatively assessed randomly. The extent of acute renal tubular necrosis was classified and scored by a semi-quantitative method: 0 points = normal, no necrosis, 1 point = 10%, 2 points = 10–25%, 3 points = 26–75%, 4 points = 75%.

### 4.6. Immunofluorescence

Immunofluorescence for lotus tetragonolobus lectin (LTL) and pERK was performed on the kidney tissues. The dewaxed and rehydrated paraffin sections (4 μm) of the kidney tissues were washed in phosphate-buffered saline (PBS) and were stained with the primary antibodies against LTL (1:250) and pERK 1/2 (1:250) at 4 °C overnight, followed by the fluorescence-labeled secondary antibodies Alexa Fluor 488 goat anti-rabbit (1:2000; A27034; Invitrogen; Thermo Fisher Scientific, Waltham, MA, USA) at room temperature for 2 h. The nuclei were counterstained with 4′,6-diamidino-2-phenylindole (DAPI, 5 ng/mL) at room temperature for 5 min. The stained sections were visualized using a confocal laser microscope (Olympus Corporation, Tokyo, Japan).

### 4.7. TUNEL Assay

Terminal deoxynucleotidyl transferase dUTP nick-end labeling (TUNEL) assay was performed according to the manufacturer’s instructions (Roche, Basel, Switzerland) to detect apoptosis. The dewaxed and rehydrated paraffin sections (4 μm) of the kidney tissues were washed in phosphate-buffered saline (PBS) and digested for 20 min at 37 °C with 3 g/mL proteinase K. After rinsing in PBS, they were incubated in TUNEL reaction mixture at 37 °C for 60 min. Images of the tissue were photographed for analysis using a confocal laser microscope (Olympus Corporation, Tokyo, Japan).

### 4.8. Real-Time RT-PCR

Total RNA was isolated from kidney using the Easy-Blue RNA extraction kit. RNA purity was confirmed by a Gene Quant Pro RNA calculator (Biochrom Inc., Holliston, MA, USA). Reverse transcription of RNA to cDNA was performed using a ABI cDNA synthesis kit (Applied Biosystems; Thermo Fisher Scientific, Waltham, MA, USA). TaqMan quantitative RT-PCR was performed using an ABI StepOne Plus detection system according to the manufacturer’s instructions. For each sample, triplicate test reactions and a control reaction without reverse transcriptase were analyzed to evaluate the expression of the gene of interest and control the variations in the reactions. The mRNA levels of the genes were normalized against those of the housekeeping gene, hypoxanthine-guanine phosphoribosyl-transferase (HPRT). qPCR was performed at 50 °C for 2 min and 95 °C for 10 min, followed by 60 cycles of amplification at 95 °C for 10 s and 60 °C for 30 s. Forward, reverse, and probe oligonucleotide primers for multiplex real-time TaqMan PCR were purchased from Applied Biosystems. The forward, reverse, and probe oligonucleotide primers for multiplex real-time TaqMan PCR were as follows: For mouse TNF-α forward, 5’-TCT CTT CAA GGG ACA AGG CTG-3’, reverse, 5’-ATA GCA AAT CGG CTG ACG GT-3’; mouse IL-1β forward, 5’-TTG ACG GAC CCC AAA AGA T-3’, reverse, 5’-GAA GCT GGA TGC TCT CAT CTG-3’; mouse IL-6 forward, 5’-TTC ATT CTC TTT GCT CTT GAA TTA GA-3’, reverse, 5’-GTC TGA CCT TTA GCT TCA AAT CCT-3’; mouse HPRT (forward, 5’-GAC CGG TCC CGT CAT GC-3’; reverse, 5’-CAT AAC CTG GTT CAT CAT CGC TAA-3’).

### 4.9. Western Blot

Renal tissues were homogenized and lysed in ice. The lysates were boiled in 62.5 mM Tris-HCl buffer with a pH of 6.8 that contained 2% sodium dodecyl sulfate (SDS), 20% glycerol, and 10% 2-mercaptoethanol. Proteins were separated on a 10% SDS-polyacrylamide gel and transferred to a nitrocellulose membrane. The membrane was blocked with 5% skim milk in PBS with Tween 20 (PBST) for 2 h under room temperature (RT), followed by incubation with primary antibodies overnight at 4 °C. After washing three times, the membrane was incubated with secondary antibodies for 1 h under RT. The proteins were visualized using an enhanced chemiluminescence detection system (Amersham, Buckinghamshire, UK) according to the manufacturer’s recommended protocol.

### 4.10. Statistical Analysis

Results are expressed as mean ± standard error of the mean (SEM). The significance of the differences was evaluated using one-way analysis of variance (ANOVA) and post hoc tests were performed using Duncan’s procedure. All the statistical analyses were performed using SPSS statistical analysis software v. 10.0 (SPSS Inc., Chicago, IL, USA). Values of *p* < 0.05 were considered to be statistically significant.

## 5. Conclusions

In summary, we demonstrated that loganin exhibited reno-protective effects against cisplatin-induced AKI via the deactivation of ERK 1/2. Taken together, our results suggest that loganin may be an effective adjuvant for cisplatin-based cancer therapy. Further study would be needed to evaluate the synergistic or therapeutic effects of loganin in cisplatin-based cancer treatment.

## Figures and Tables

**Figure 1 ijms-22-01421-f001:**
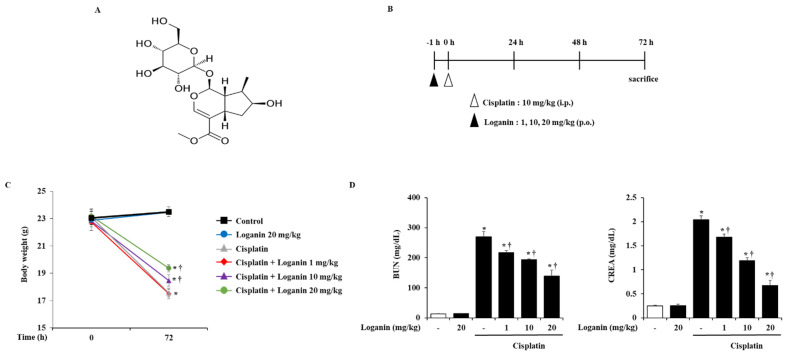
Effect of loganin on blood urea nitrogen (BUN) and creatinine (CREA) in cisplatin-induced acute kidney injury (AKI). (**A**) Molecular structure of loganin. (**B**) Mice pretreated with loganin (1, 10, or 20 mg/kg) were administered with intraperitoneal injections of cisplatin (10 mg/kg). They were killed at 72 h after the cisplatin injection. (**C**) Body weight changes, (**D**) serum BUN and CREA were measured at 72 h after cisplatin injection. Each experiment was repeated 3 times. Data are represented as mean ± SEM (*n* = 9). An * indicates *p* < 0.05 vs. saline-treated control group; † indicates *p* < 0.05 vs. cisplatin treatment alone.

**Figure 2 ijms-22-01421-f002:**
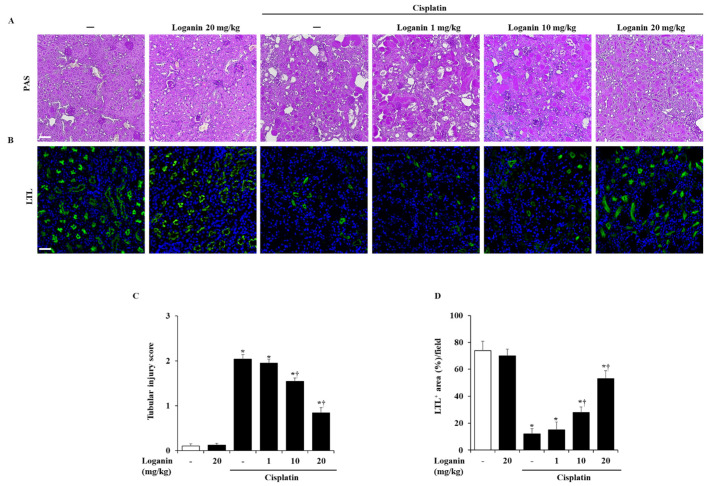
Effect of loganin on renal histologic changes in cisplatin-induced AKI. (**A**) Representative periodic acid–Schiff (PAS)-stained sections of the kidney (200 × magnification). (**B**) Representative lotus tetragonolobus lectin (LTL)-stained sections of the kidney (200 × magnification). (**C**) Histological sections of the kidney were scored from 0 (normal) to 4 (severe) for necrosis of tubule cells. (**D**) LTL-positive areas of the kidney were scored for green fluorescence. Each experiment was repeated 3 times. Data are represented as mean ± SEM (*n* = 9). An * indicates *p* < 0.05 vs. saline-treated control group; † indicates *p* < 0.05 vs. cisplatin treatment alone. Scale bar = 20 μm.

**Figure 3 ijms-22-01421-f003:**
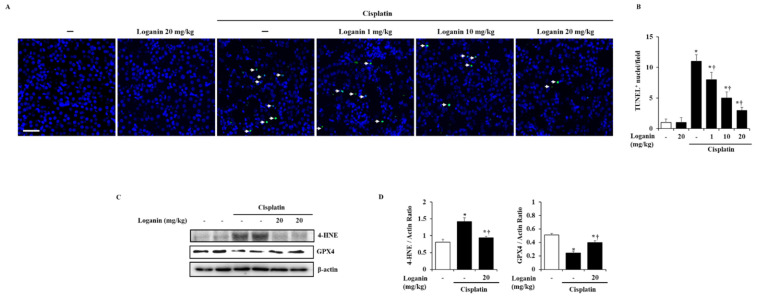
Effect of loganin on renal tubular apoptosis and ferroptosis in cisplatin-induced AKI. (**A**) Representative terminal deoxynucleotidyl transferase dUTP nick-end labeling (TUNEL)-stained sections of the kidney (600 × magnification). (**B**) TUNEL-positive cells (indicated by arrow) of the kidney were scored for green fluorescence. (**C**) Anti-4-hydroxynonenal (4-HNE) and glutathione peroxidase 4 (GPX4) were analyzed by Western blot. β-actin was used as a loading control. (**D**) The relative density ratio of 4-HNE/β-actin and GPX4/β-actin. Each experiment was repeated 3 times. Data are represented as mean ± SEM (*n* = 9). An * indicates *p* < 0.05 vs. saline-treated control group; † indicates *p* < 0.05 vs. cisplatin treatment alone. Scale bar = 10 μm.

**Figure 4 ijms-22-01421-f004:**
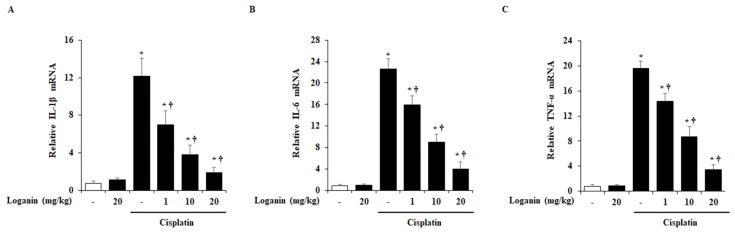
Effect of loganin on mRNA levels of pro-inflammatory cytokine in cisplatin-induced AKI. mRNA levels of (**A**) interleukin (IL)-1β, (**B**) IL-6, and (**C**) tumor necrosis factor (TNF)-α were detected by real-time PCR. Each experiment was repeated 3 times. Data are represented as mean ± SEM (*n* = 9). An * indicates *p* < 0.05 vs. saline-treated control group; † indicates *p* < 0.05 vs. cisplatin treatment alone.

**Figure 5 ijms-22-01421-f005:**
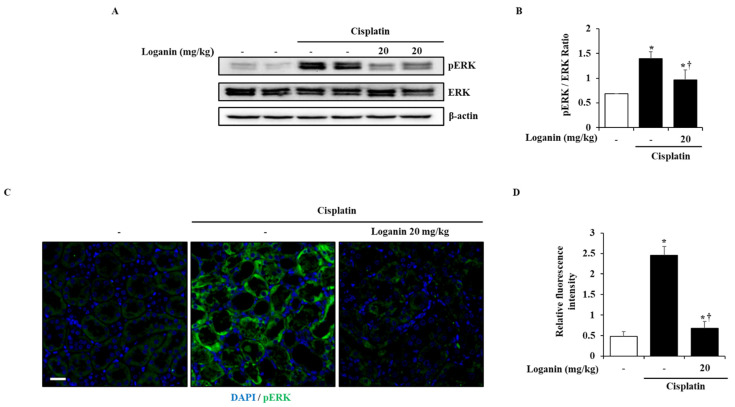
Effect of loganin on phosphorylation of extracellular signal-regulated kinases (ERK) 1/2 in cisplatin-induced AKI. (**A**) The phosphorylation of ERK 1/2 was analyzed by Western blot. Total ERK1/2, and β-actin were used as loading controls. (**B**) The relative density ratio of pERK1/2/ERK1/2. (**C**) Representative pERK1/2-stained sections of the kidney (200 × magnification). (**D**) pERK1/2-positive areas of the kidney were scored for green fluorescence. Each experiment was repeated 3 times. Data are represented as mean ± SEM (*n* = 9). An * indicates *p* < 0.05 vs. saline-treated control group; † indicates *p* < 0.05 vs. cisplatin treatment alone. Scale bar = 20 μm.

**Figure 6 ijms-22-01421-f006:**
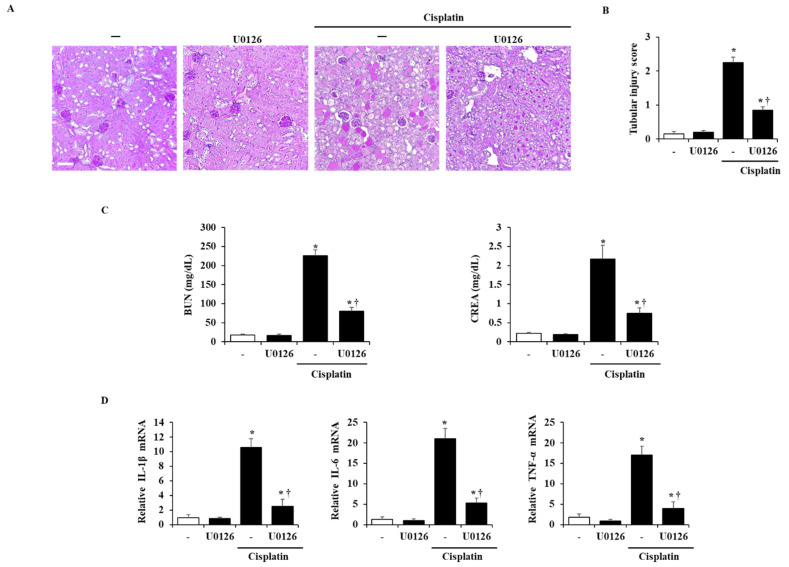
Effects of ERK 1/2 on the severity of cisplatin-induced AKI. Mice pretreated with U0126, ERK 1/2 inhibitor, (10 mg/kg) were administered with intraperitoneal injections of cisplatin (10 mg/kg). They were killed at 72 h after the cisplatin injection. (**A**) Representative PAS-stained sections of the kidney (200× magnification). (**B**) Histological sections of the kidney were scored from 0 (normal) to 4 (severe) for necrosis of tubule cells. (**C**) Serum BUN and CREA were measured at 72 h after cisplatin injection. (**D**) mRNA levels of IL-1β, IL-6, and TNF-α were detected by real-time PCR. Each experiment was repeated 3 times. Data are represented as mean ± SEM (*n* = 9). An * indicates *p* < 0.05 vs. saline-treated control group; † indicates *p* < 0.05 vs. cisplatin treatment alone. Scale bar = 20 μm.

**Table 1 ijms-22-01421-t001:** Effects of loganin on body weight in cisplatin-induced AKI.

Group	Body Weight (Day 0, g)	Body Weight (Day 3, g)	Body Weight Gain (g)
N	22.88 ± 0.48	23.46 ± 0.36	0.58 ± 0.20
Lo 20mg/kg	23.05 ± 0.46	23.50 ± 0.39	0.45 ± 0.15
Cis	23.13 ± 0.58	17.56 ± 0.29 *	−5.57 ± 0.41 *
Cis + Lo 1 mg/kg	22.75 ± 0.34	17.51 ± 0.38 *	−5.24 ± 0.30 *
Cis + Lo 10 mg/kg	22.83 ± 0.68	18.46 ± 0.42 *^,†^	−4.37 ± 0.28 *^,†^
Cis + Lo 20 mg/kg	23.20 ± 0.44	19.36 ± 0.25 *^,†^	−3.84 ± 0.21 *^,†^

* indicates *p* < 0.05 vs. saline-treated control group, ^†^ indicates *p* < 0.05 vs. cisplatin treatment alone. N; Normal, Lo; Loganin, Cis; Cisplatin.

## Data Availability

The datasets used and/or analyzed during the current study are available from the corresponding author on reasonable request.

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
