# Peer review of "Loganin Attenuates the Severity of Acute Kidney Injury Induced by Cisplatin through the Inhibition of ERK Activation in Mice"

_ijms, 2021, doi:10.3390/ijms22031421_

Round 1

Reviewer 1 Report

In this work the authors decribe the possibility of using Loganin to attenuate the severity of acute liver injury induced by cisplatin.

Overall the work is well-designed and written. Only minor points should be addressed:

  1. Please provide the molecular structure of Loganin in the context of a figure showing where it could act. For instance pertinent enzyme(s) should be presented in an appropriate cascade with a putative role of this compound in altering the intra-cellular events. The biochemistry is not clear and not widely understood.
  2. The title is in "past tense". Wouldn't it be better to use present tense? This way it would imply a general mechanism, rather than a recorded event.
  3. Figure 1B. Weight gain with negative values is counterintuitive.
  4. Lines 254-256 could be integrated and yield a separate "Conclusion" section of the study 

Author Response

Q1. Please provide the molecular structure of Loganin in the context of a figure showing where it could act. For instance pertinent enzyme(s) should be presented in an appropriate cascade with a putative role of this compound in altering the intra-cellular events. The biochemistry is not clear and not widely understood.

A1. Thanks for your good indication. As you suggested, we added the molecular structure of loganin in Figure 1A. 

Q2. The title is in "past tense". Wouldn't it be better to use present tense? This way it would imply a general mechanism, rather than a recorded event.

 A2. Thanks for your good indication. We revised the title as follows.

Loganin attenuates the severity of acute kidney in-jury induced by cisplatin through the inhibition of ERK activation in mice.

Q3. Figure 1B. Weight gain with negative values is counterintuitive. 

A3. Thanks for your good indication. We modified the figure so that the body weight change from day 0 to day 3 can be seen at a glance in Figure 1C.

Q4. Lines 254-256 could be integrated and yield a separate "Conclusion" section of the study.

A4. Thanks for your good indication. We have separated those sentences into a "conclusion" section in page 8.

4. Conclusion

In summary, we have demonstrated that loganin exhibited the reno-protective effects against cisplatin-induced AKI via the deactivation of ERK 1/2. Taken together, our results suggest that loganin may be an effective adjuvant for cisplatin-based cancer therapy. The further study would be needed to evaluate the synergistic or therapeutic effects of loganin in the cisplatin-based cancer treatment.

Reviewer 2 Report

Dear Authors,

the manuscript "Loganin attenuated the severity of acute kidney injury induced by cisplatin through the inhibition of ERK activation in mice" describes the effect of the Corni fructus on AKI induced by cisplatin. The manuscript has merit, however there are some points that need explanation:

  1. Loganin can be used as a potential drug - when the compound will be used together in the treatment of cancers with cisplatin, then the effect of cisplatin on cancer cells will be insufficient. Could the Authors explain this?
  2. Figure 3 - please check if the photo is correct desribed (the length of the line above the picture isn't too long?)
  3. Why did not the Authors show the effect of loganin alone on 4-HNE and GPX4 (Fig. 3C, D)? The same is in Fig. 5 A-D.
  4. Lines 132-133 - why these data are not shown? The Authors should present these results, especially that they discuss about them.   

Author Response

Q1. Loganin can be used as a potential drug - when the compound will be used together in the treatment of cancers with cisplatin, then the effect of cisplatin on cancer cells will be insufficient. Could the Authors explain this?

A1. Thanks for your sharp indication. As you indicated, when the loganin would be used together in the treatment of cancers, it is so important to explain whether loganin might affect the anti-tumor effects of cisplatin, because killing cancer cells is a main role of cisplatin even with the minor side effects such acute kidney injury (AKI). Thus we searched the published reports about anti-cancer effects of loganin, but failed to find any reports about anti-cancer in Pubmed. However, we could find the pertainable paper to this question, which demonstrate that loganetin (the metabolites of loganin [1]), and 5-fluorouracil (5-FU, anti-cancer agents) synergistically inhibit the carcinogenesis [2]. In that paper, loganetin could not only efficiently inhibit gastric cancer cells alone, but also synergistically enhanced the anti-tumor effect of 5-FU. These results could suggest that loganin could be used in cancer therapy alone as well as combination with chemotherapeutic agents (5-FU). Although the chemotherapeutic agent (5-FU) is different from cisplatin, we could suppose that loganin/loganetin (the metabolite of loganin) would be helpful in the treatment of cancers. Therefore, if loganin is properly used with cisplatin in the treatment of cancers, the nephrotoxicity of cisplatin would not only be reduced, but also the anti-cancer effect of cisplatin may be enhanced. The further study will have to be conducted for the evaluation the synergistic or therapeutic effects of loganin in the cisplatin-based cancer treatment.

We added these comments in introduction (2nd paragraph, Introduction in page 2) and conclusion (Conclusion in page 8).

Q2. Figure 3 - please check if the photo is correct described (the length of the line above the picture isn't too long?)

A2. Thanks for your sharp indication and sorry for the confusion. We checked the figures again and we found the same problem with Figure 2. So we revised the length of the lines in both figures.

Q3. Why did not the Authors show the effect of loganin alone on 4-HNE and GPX4 (Fig. 3C, D)? The same is in Fig. 5 A-D.

A3. Thanks for your good indication. According to the results of the experiment (body weight change, serum levels of blood urea nitrogen and creatinine, histological kidney changes, and inflammatory cytokine levels), loganin alone group did not show any difference from the normal control group. Thus, we concluded that loganin alone might not show any activities in cell death (Figure 3), and inflammatory mechanisms (Figure 5), and did not examine the loganin alone sample for Western blot experiments (4-HNE, GPX4, pERK 1/2). Please give us big generosity about our performances.

Q4. Lines 132-133 - why these data are not shown? The Authors should present these results, especially that they discuss about them. 

A4. Thanks for your good indication. There are obvious mistakes during English correction and re-writing. We committed an error in marking the parentheses [⊃] in the middle of the sentences. We intended to state that administration of loganin DO NOT inhibit the degradation of Iκ-Bα. So, we modified the sentence as follows.

Original manuscripts

However, the administration of loganin inhibited the phosphorylation of ERK 1/2 (not p38 and JNK) and the degradation of Iκ-Bα (Figure 5A and data not shown).

Revised manuscript

However, the administration of loganin inhibited the phosphorylation of ERK 1/2 only, but not the phosphorylation p38 and JNK and the degradation of Iκ-Bα (Fig. 5A and B and Supplementary Figure S2).

In addition, we added the results of negative data (pp38, pJNK, Iκ-Bα) with immunostaining in Supplementary Figure S2) as below.

References

  1. Zhao, M.; Xu, J.; Qian, D.; Guo, J.; Jiang, S.; Shang, E.X.; Duan, J.A.; Du, L. Characterization of in Vitro Metabolism of Loganin by Human Intestinal Microflora Using Ultra-High Performance Liquid Chromatography–Quadrupole Time-of-Flight Mass Spectrometry. Anal Lett 2014, 47, 1500-1512.
  2. Zhou, H.; Hu, X.; Li, N.; Li, G.; Sun, X.; Ge, F.; Jiang, J.; Yao, J.; Huang, D.; Yang, L. Loganetin and 5-fluorouracil synergistically inhibit the carcinogenesis of gastric cancer cells via down-regulation of the Wnt/β-catenin pathway. J Cell Mol Med 2020, 24, 13715-13726.

Round 2

Reviewer 2 Report

Dear Authors,

my sugestions have been improved. The manuscript is now suitable for publication.